# SGLang: Efficient Execution of Structured Language Model Programs

**Lianmin Zheng**[2*]   **Liangsheng Yin**[3]   **Zhiqiang Xie**[1]   **Chuyue Sun**[1]   **Jeff Huang**[4]
**Cody Hao Yu**[5]   **Shiyi Cao**[2]   **Christos Kozyrakis**[1]   **Ion Stoica**[2]   **Joseph E. Gonzalez**[2]
**Clark Barrett**[1]   **Ying Sheng**[1*]

[1] Stanford University   [2] UC Berkeley   [3] Shanghai Jiao Tong University
[4] Texas A&M University   [5] Independent Researcher

## Abstract

Large language models (LLMs) are increasingly used for complex tasks that require multiple generation calls, advanced prompting techniques, control flow, and structured inputs/outputs. However, efficient systems are lacking for programming and executing these applications. We introduce SGLang, a system for efficient execution of complex language model programs. SGLang consists of a frontend language and a runtime. The frontend simplifies programming with primitives for generation and parallelism control. The runtime accelerates execution with novel optimizations like RadixAttention for KV cache reuse and compressed finite state machines for faster structured output decoding. Experiments show that SGLang achieves up to $6.4\times$ higher throughput compared to state-of-the-art inference systems on various large language and multi-modal models on tasks including agent control, logical reasoning, few-shot learning benchmarks, JSON decoding, retrieval-augmented generation pipelines, and multi-turn chat. The code is publicly available at https://github.com/sgl-project/sglang.

## 1   Introduction

Recent increases in the capabilities of LLMs have broadened their utility, enabling them to tackle a wider range of general tasks and act as autonomous agents [35, 6, 36, 52, 46]. In such applications, LLMs engage in multi-round planning, reasoning, and interaction with external environments. This is accomplished through tool usage [41, 38], multiple input modalities [47, 2], and a wide range of prompting techniques [30], like few-shot learning [5], self-consistency [53], skeleton-of-thought [33], and tree-of-thought [56]. All of these new use cases require multiple, often dependent, LLM generation calls, showing a trend of using multi-call structures to complete complex tasks [57, 21].

The emergence of these patterns signifies a shift in our interaction with LLMs, moving from simple chatting to a more sophisticated form of programmatic usage of LLMs, which means using a program to schedule and control the generation processes of LLMs. We refer to these programs as "Language Model Programs" (LM Programs) [4, 20]. The advanced prompting techniques and agentic workflow mentioned above fall within the scope of LM programs. There are two common properties of LM programs: (1) LM programs typically contain multiple LLM calls interspersed with control flow. This is needed to complete complex tasks and improve overall quality. (2) LM programs receive structured inputs and produce structured outputs. This is needed to enable the composition of LM programs and to integrate LM programs into existing software systems.

---

*Equal contribution.

38th Conference on Neural Information Processing Systems (NeurIPS 2024).

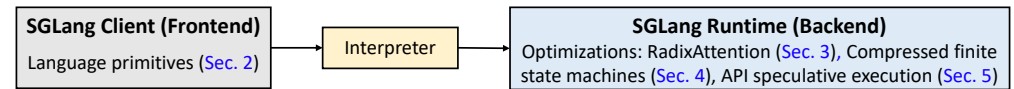

Figure 1: System architecture: An interpreter executes language primitives with optimized runtime.

Despite the widespread use of LM programs, current systems for expressing and executing them remain inefficient. We identify two primary challenges associated with the efficient use of LM programs: *First, programming LM programs is tedious and difficult due to the non-deterministic nature of LLMs.* Developing an LM program often requires extensive string manipulation, experimental tuning of prompts, brittle output parsing, handling multiple input modalities, and implementing parallelism mechanisms. This complexity significantly reduces the readability of even simple programs (Sec. 2).

*Secondly and importantly, executing LM programs is inefficient due to redundant computation and memory usage.* State-of-the-art inference engines (e.g., vLLM [23], TGI [16], and TensorRT-LLM [34]), have been optimized to reduce latency and improve throughput without direct knowledge of the workload. This makes these systems general and robust but also results in significant inefficiencies for any given workload. A prominent example is the reuse of the Key-Value (KV) cache (Sec. 3). The KV cache consists of reusable intermediate tensors that are essential for generative inference. During typical batch executions of LM programs, numerous opportunities exist to reuse the KV cache across multiple different LLM calls that share a common prefix. However, current systems lack effective mechanisms to facilitate this reuse, resulting in unnecessary computations and wasted memory. Another example is constrained decoding for structured outputs (e.g., JSON mode), where the output of LLMs is restricted to follow specific grammatical rules defined by a regular expression (Sec. 4). Under these constraints, multiple tokens can often be decoded once. However, existing systems only decode one token at a time, leading to suboptimal decoding speeds.

To address these challenges, we present SGLang, a Structured Generation Language for LLMs. The core idea is to systematically exploit the multi-call structure in LM programs for efficient execution. As shown in Fig. 1, it has two parts: a front-end language and a back-end runtime. The front-end simplifies the programming of LM programs, and the runtime accelerates their execution. The two parts can work together for better performance but can also function independently.

We introduce SGLang as a domain-specific language embedded in Python. It provides primitives for generation (e.g., extend, gen, select) and parallelism control (e.g., fork, join). SGLang is compatible with Python's control flow and libraries, so users can develop advanced prompting workflows easily with native Python syntax. We provide an interpreter and a compiler for SGLang. The interpreter manages the prompt state as a stream and submits primitive operations to the stream for asynchronous execution, ensuring proper control over synchronization and intra-program parallelism. Additionally, SGLang program can be traced and compiled for more optimizations.

On the runtime side, we propose several novel optimizations to accelerate the execution of SGLang programs. *The first technique*, RadixAttention, enables the automatic reuse of the KV cache across multiple generation calls. In existing inference engines, the KV cache of a request is discarded after processing is completed, preventing the KV cache from being reused across multiple calls and significantly slowing down the execution. Instead, our system maintains an LRU cache of the KV cache for all requests within a radix tree. This approach manages the KV cache as a traditional cache and uses a radix tree for efficient matching, insertion, and eviction. It allows the runtime to handle various reuse patterns with a cache-aware scheduling policy efficiently. *The second technique* is a compressed finite state machine, which enables faster constrained decoding for structured outputs. Existing systems follow the constraints only for the next token by masking probabilities of disallowed tokens, making them able to decode only one token at a time. Instead, our system analyzes the constraints and builds a compressed finite-state machine to represent the constraint. This approach compresses a multi-token path into a single-step path whenever possible, allowing the decoding of multiple tokens at once to achieve faster decoding speed. Lastly, SGLang also supports API-only models like OpenAI's GPT-4, and we introduce *the third technique*, API speculative execution, to optimize multi-call programs for API-only models.

Using SGLang, we implemented various LLM applications, including agent control, logical reasoning, few-shot learning benchmarks, JSON decoding, retrieval-augmented generation pipelines, multi-turn chat, and multi-modality processing. We tested the performance on models including Llama-7B/70B [49], Mistral-8x7B [17], LLaVA-v1.5-7B (image) [28], and LLaVA-NeXT-34B (video) [62] on NVIDIA A10G and A100 GPUs. Experimental results show that SGLang achieves up to $6.4\times$ higher throughput across a wide range of workloads, models, and hardware setups, compared to existing programming and inference systems, including Guidance [13], vLLM [23], and LMQL [4].

```python
dimensions = ["Clarity", "Originality", "Evidence"]
@function
def multi_dimensional_judge(s, path, essay):
  s += system("Evaluate an essay about an image.")
  s += user(image(path) + "Essay:" + essay)          # Handle chat template
  s += assistant("Sure!")                              # and multi-modal inputs

  # Return directly if it is not related             # Select an option with
  s += user("Is the essay related to the image?")     # the highest probability
  s += assistant(select("related", choices=["yes", "no"]))
  if s["related"] == "no": return                      # Fetch result; Use Python
                                                       # control flow
  # Judge multiple dimensions in parallel
  forks = s.fork(len(dimensions))                      # Runtime optimization:
  for f, dim in zip(forks, dimensions):                # KV Cache Reuse (Sec. 3)
    f += user("Evaluate based on the following dimension:" +
      dim + ". End your judgment with the word 'END'")
    f += assistant("Judgment:" + gen("judgment", stop="END"))   # Multiple generation
                                                                 # calls run in parallel
  # Merge the judgments
  judgment = "\n".join(f["judgment"] for f in forks)   # Fetch generation results

  # Generate a summary and a grade. Return in the JSON format.
  s += user("Provide the judgment, summary, and a letter grade")
  s += assistant(judgment + "In summary," + gen("summary", stop=".")   # Runtime optimization: API
                 + "The grade of it is" + gen("grade"))               # speculative execution (Sec. 5)
  schema = r'\{"summary": "[\w\d\s]+\.", "grade": "[ABCD][+-]?"\}'
  s += user("Return in the JSON format.")
  s += assistant(gen("output", regex=schema))          # Runtime optimization: fast
                                                        # constrained decoding (Sec. 4)
state = multi_dimensional_judge.run(...)
print(state["output"])                                 # Run an SGLang program
```

Figure 2: The implementation of a multi-dimensional essay judge in SGLang utilizes the branch-solve-merge prompting technique [40]. Primitives provided by SGLang are shown in red.

## 2 Programming Model

This section introduces the SGLang programming model with a running example, describes its language primitives and execution modes, and outlines runtime optimization opportunities. This programming model can simplify tedious operations in multi-call workflows (e.g., string manipulation, API calling, constraint specification, parallelism) by providing flexible and composable primitives.

**A running example.** The language is a domain-specific language embedded in Python. Fig. 2 shows a program that evaluates an essay about an image using the branch-solve-merge prompting method [40]. The function `multi_dimensional_judge` takes three arguments: `s`, `path`, and `essay`. `s` manages the prompt state, `path` is the image file path, and `essay` is the essay text. New strings and SGLang primitives can be appended to the state `s` for execution using the `+=` operator. First, the function adds the image and essay to the prompt. It then checks if the essay is related to the image using `select`, storing the result in `s["related"]`. If related, the prompt is forked into three copies for parallel evaluation from different dimensions, using `gen` to store results in `f["judgment"]`. Next, it merges the judgments, generates a summary, and assigns a letter grade. Finally, it returns the results in JSON format, following a schema defined by a regular expression constraint `regex`. SGLang greatly simplifies this program, as an equivalent program using an OpenAI API-like interface would take $2.1\times$ as many lines of code due to manual string manipulation and parallelism control.

**Language primitives.** SGLang provides primitives for controlling prompt state, generation, and parallelism. They can be used together with Python syntax and libraries. Here are the primitives: "`gen`" calls a model to generate and stores the results in a variable with the name specified in its first argument. It supports a "`regex`" argument to constrain the output to follow a grammar defined by a regular expression (e.g., a JSON schema). "`select`" calls a model to choose the highest probability option from a list. The operator "`+=`" or "`extend`" appends a string to the prompt. The operator "`[variable_name]`" fetches the results of a generation. "`fork`" creates parallel forks of the prompt state. "`join`" rejoins the prompt state. "`image`" and "`video`" take in image and video inputs.

**Execution modes.** The simplest way to execute an SGLang program is through an interpreter, where a prompt is treated as an asynchronous stream. Primitives like `extend`, `gen`, and `select` are submitted to the stream for asynchronous execution. These non-blocking calls allow Python code to continue running without waiting for the generation to finish. This is similar to launching CUDA kernels asynchronously. Each prompt is managed by a stream executor in a background thread, enabling intra-program parallelism. Fetching generation results will block until they are ready, ensuring correct synchronization. Alternatively, SGLang programs can be compiled as computational graphs and

Table 1: Comparison among LMQL, Guidance, and SGLang.

| System | Syntax | Language Primitives | Runtime Backends |
|--------|--------|---------------------|------------------|
| LMQL | Custom | extend, gen, select | HF Transformers, llama.cpp, OpenAI |
| Guidance | Python | extend, gen, select, image | HF Transformers, llama.cpp, OpenAI |
| SGLang | Python | extend, gen, select, image, video, fork, join | SGLang Runtime (SRT), OpenAI |

executed with a graph executor, allowing for more optimizations. This paper uses interpreter mode by default and discusses compiler mode results in Appendix D. SGLang supports open-weight models with its own SGLang Runtime (SRT), as well as API models such as OpenAI and Anthropic models.

**Comparison.** Programming systems for LLMs can be classified as high-level (e.g., LangChain, DSPy) and low-level (e.g., LMQL, Guidance, SGLang). High-level systems provide predefined or auto-generated prompts, such as DSPy's prompt optimizer. Low-level systems typically do not alter prompts but allow direct manipulation of prompts and primitives. SGLang is a low-level system similar to LMQL and Guidance. Table 1 compares their features. SGLang focuses more on runtime efficiency and comes with its own co-designed runtime, allowing for novel optimizations introduced later. High-level languages (e.g., DSPy) can be compiled to low-level languages (e.g., SGLang). We demonstrate the integration of SGLang as a backend in DSPy for better runtime efficiency in Sec. 6.

**Runtime optimizations.** Fig. 2 shows three runtime optimization opportunities: KV cache reuse, fast constrained decoding, API speculative execution. We will discuss them in the following sections.

## 3   Efficient KV Cache Reuse with RadixAttention

SGLang programs can chain multiple generation calls and create parallel copies with the "fork" primitive. Additionally, different program instances often share some common parts (e.g., system prompts). These scenarios create many shared prompt prefixes during execution, leading to numerous opportunities for reusing the KV cache. During LLM inference, the KV cache stores intermediate tensors from the forward pass, reused for decoding future tokens. They are named after key-value pairs in the self-attention mechanism [51]. KV cache computation depends only on prefix tokens. Therefore, requests with the same prompt prefix can reuse the KV cache, reducing redundant computation and memory usage. More background and some examples are provided in Appendix A.

Given the KV cache reuse opportunity, a key challenge in optimizing SGLang programs is reusing the KV cache across multiple calls and instances. While some systems explore certain KV cache reuse cases [23, 58, 18, 12], they often need manual configurations and cannot handle all reuse patterns (e.g., dynamic tree structures). Consequently, most state-of-the-art inference systems recompute the KV cache for each request. We will discuss their limitations and our differences in Sec. 7.

This section introduces RadixAttention, a novel technique for automatic and systematic KV cache reuse during runtime. Unlike existing systems that discard the KV cache after a generation request finishes, our system retains the cache for prompts and generation results in a radix tree, enabling efficient prefix search, reuse, insertion, and eviction. We implement an LRU eviction policy and a cache-aware scheduling policy to enhance the cache hit rate. RadixAttention is compatible with techniques like continuous batching [60], paged attention [23], and tensor parallelism [44]. In addition, it introduces only negligible memory and time overhead when there is no cache hit.

**RadixAttention.** A radix tree is a data structure that serves as a space-efficient alternative to a classical trie (prefix tree). Unlike typical trees, the edges of a radix tree can be labeled not just with single elements but also with sequences of elements of varying lengths, significantly enhancing efficiency. In our system, we utilize a radix tree to manage a mapping between sequences of tokens, and their corresponding KV cache tensors. These KV cache tensors are stored in a non-contiguous, paged layout, where the size of each page is equivalent to one token. Because GPU memory is quickly filled by the KV cahce, we introduce a simple LRU eviction policy that evicts the least recently used leaf first. By evicting leaves first, we enable the re-use of their common ancestors until those ancestors become leaves and are also evicted.

In the continuous batching setting, we cannot evict nodes used by the currently running batch. Therefore, each node maintains a reference counter indicating how many running requests are using it. A node is evictable if its reference counter is zero. Note that we do not preallocate a fixed-size memory pool as a cache. Instead, we let the cached tokens and the currently running requests share the same memory pool. Therefore, the system dynamically allocates memory for cache and running requests. When enough waiting requests run, the system will evict all cached tokens in favor of a larger batch size. Fig. 3 shows how the radix tree is maintained for several incoming requests. The

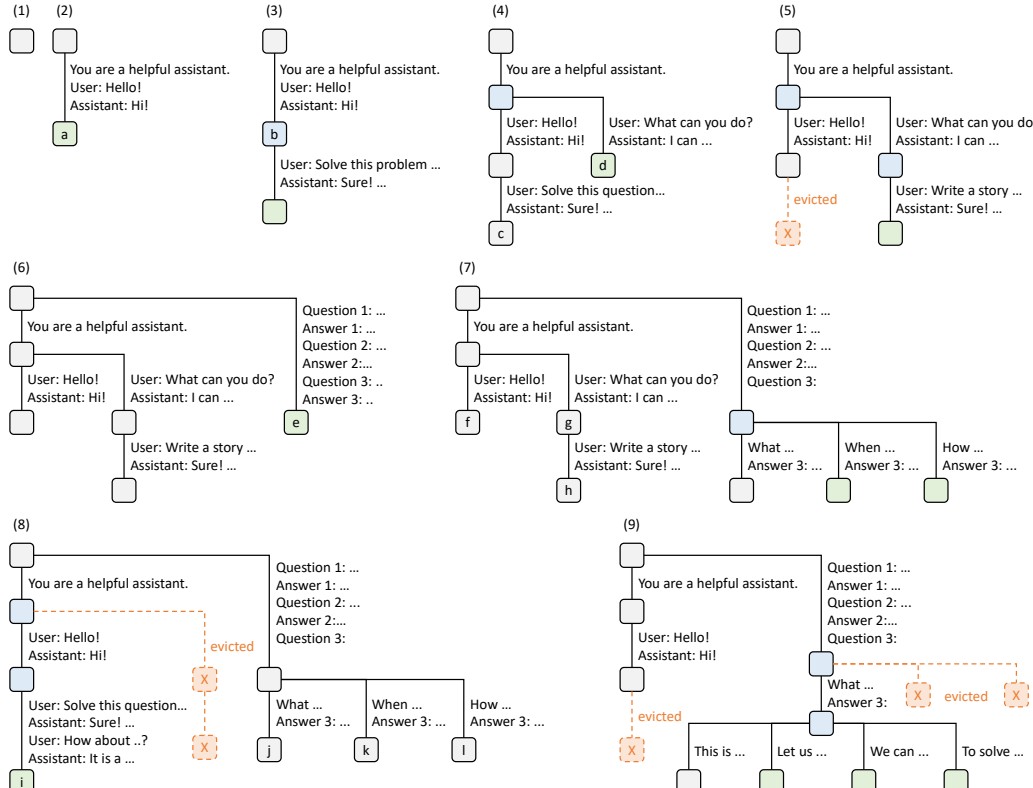

Figure 3: Examples of RadixAttention operations with an LRU eviction policy, illustrated across nine time points. The figure demonstrates the dynamic evolution of the radix tree in response to various requests. These requests include two chat sessions, a batch of few-shot learning inquiries, and a self-consistency sampling. Each tree edge carries a label denoting a substring or a sequence of tokens. The nodes are color-coded to reflect different states: green for newly added nodes, blue for cached nodes accessed during the time point, and red for nodes that have been evicted. In step (1), the radix tree is initially empty. In step (2), the server processes an incoming user message "Hello" and responds with the LLM output "Hi". The system prompt "You are a helpful assistant", the user message "Hello!", and the LLM reply "Hi!" are consolidated into the tree as a single edge linked to a new node. In step (3), a new prompt arrives and the server finds the prefix of the prompt (i.e., the first turn of the conversation) in the radix tree and reuses its KV cache. The new turn is appended to the tree as a new node. In step (4), a new chat session begins. The node "b" from (3) is split into two nodes to allow the two chat sessions to share the system prompt. In step (5), the second chat session continues. However, due to the memory limit, node "c" from (4) must be evicted. The new turn is appended after node "d" in (4). In step (6), the server receives a few-shot learning query, processes it, and inserts it into the tree. The root node is split because the new query does not share any prefix with existing nodes. In step (7), the server receives a batch of additional few-shot learning queries. These queries share the same set of few-shot examples, so we split node 'e' from (6) to enable sharing. In step (8), the server receives a new message from the first chat session. It evicts all nodes from the second chat session (node "g" and "h") as they are least recently used. In step (9), the server receives a request to sample more answers for the questions in node "j" from (8), likely for self-consistency prompting. To make space for these requests, we evict node "i", "k", and "l" in (8).

frontend interpreter sends full prompts to the runtime, and the runtime performs prefix matching and reuse. The tree structure is stored on the CPU with negligible maintenance overhead. During the execution of the `fork` primitive, the frontend sends the prefix first as a hint, ensuring the prefix is correctly inserted into the tree. It then sends the remaining prompts. This "Frontend Hint" simplifies runtime scheduling and matching, exemplifying the benefits of frontend-runtime co-design.

**Cache-aware scheduling.** We define the cache hit rate as $\frac{\text{number of cached prompt tokens}}{\text{number of prompt tokens}}$. When there are many requests in the waiting queue, the order in which they are executed can significantly impact the cache hit rate. For example, if the request scheduler frequently switches between different, unrelated requests, it can lead to cache thrashing and a low hit rate. We design a cache-aware scheduling algorithm to increase the cache hit rate. In the batch-processing setting we sort the

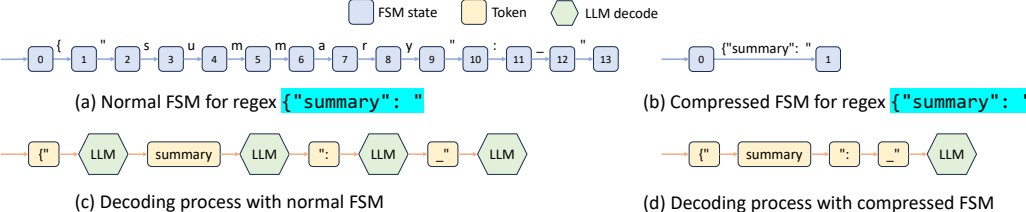

Figure 4: The decoding process of normal and compressed FSMs (the underscore _ means a space).

requests by matched prefix length and prioritize requests with longer matched prefixes instead of using a first-come, first-served schedule. Alg. 1 (Appendix) shows the pseudo-code for cache-aware scheduling with contiguous batching. The algorithm uses longest-shared-prefix-first order. In more latency-sensitive settings we may still be able to tolerate limited batch re-ordering to improve cache reuse. Additionally, we prove the following theorem for optimal scheduling in the offline case. [2]

**Theorem 3.1.** *For a batch of requests, we can achieve an optimal cache hit rate by visiting the radix tree of the requests in the depth-first search order, with a cache size $\geq$ the maximum request length. The longest-shared-prefix-first order is equivalent to a depth-first search order.*

The proof is in Sec. A.3 (Appendix). In the online case, the DFS order will be disrupted, but our schedule still approximates the DFS behavior on the augmented part of the full radix tree, as described in Sec. A.3. While greedy cache-aware scheduling can achieve high throughput, it can lead to starvation. We leave its integration with other fair scheduling methods [42] as future work.

**Distributed Cases.** RadixAttention can be extended to multiple GPUs. For tensor parallelism, each GPU maintains a sharded KV cache. There is no need for additional synchronization because the tree operations are the same. Data parallelism with multiple workers is discussed in Sec. A.4 (Appendix).

# 4 Efficient Constrained Decoding with Compressed Finite State Machine

In LM programs, users often want to constrain the model's output to follow specific formats, such as JSON schemas. This can improve controllability and robustness, and make the output easier to parse. SGLang offers a `regex` argument to enforce such constraints using regular expressions, which are expressive enough for many practical scenarios. Existing systems support this by converting a regular expression into a finite state machine (FSM) [54]. During decoding, they maintain the current FSM state, retrieve allowed tokens from the next states, and set the probability of invalid tokens to zero, decoding token by token. This token-by-token approach, however, is inefficient when there are opportunities to decode multiple tokens at once. For example, the constant sequence {"summary": " in Fig. 2 spans multiple tokens in the normal decoding process as shown in Fig. 4 (c), requiring multiple decoding stages, even though there is only one valid next token when decoding it. Therefore, the whole sequence can be decoded in a single step (i.e., forward pass). However, existing systems can only decode one token at a time because the lack of integration between the FSM and the model runner in existing systems prevents multi-token processing, resulting in slow decoding.

SGLang overcomes this limitation by creating a fast constrained decoding runtime with a compressed FSM. This runtime analyzes the FSM and compresses adjacent singular-transition edges in the FSM into single edges as demonstrated in Fig. 4 (b), allowing it to recognize when multiple tokens can be decoded together. In Fig. 4 (d), multiple tokens on the compressed transition edge can be decoded in one forward pass, which greatly accelerates the decoding process. It is also general and applicable to all regular expressions. More details on the background and implementation are in Appendix B.

# 5 Efficient Endpoint Calling with API Speculative Execution

The previous sections introduced optimizations for open-weight models, which require modifications to the model inference process. Additionally, SGLang works with API-access-only models, such as OpenAI's GPT-4. However, for these models, we can only call a black-box API endpoint.

This section introduces a new optimization for black-box API models that accelerates execution and reduces the API cost of multi-call SGLang programs using speculative execution. For example, a program may ask the model to generate a description of a character with a multi-call

---

[2]In practice, the computation is not the same as what is described in the proof of Theorem 3.1 because the unpredictable number of output tokens can cause the recomputation of the KV cache.

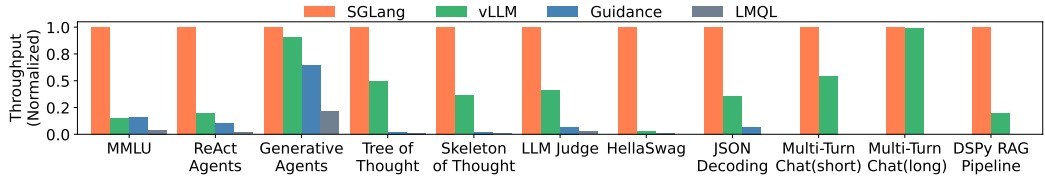

Figure 5: Normalized throughput on Llama-7B models. Higher is better.

pattern: `s += context + "name:" + gen("name", stop="\n") + "job:" + gen("job", stop="\n")`. Naively, the two `gen` primitives correspond to two API calls, meaning that the user needs to pay for the input token fee on the `context` twice. In SGLang, we can enable speculative execution on the first call and let it continue the generation of a few more tokens by ignoring the stop condition. The interpreter keeps the additional generation outputs and matches and reuses them with later primitives. In certain cases, with careful prompt engineering, the model can correctly match the template with high accuracy, saving us the latency and input costs of one API call.

# 6 Evaluation

We evaluate the performance of SGLang across diverse LLM workloads. Subsequently, we conduct ablation studies and case studies to demonstrate the effectiveness of specific components. SGLang is implemented in PyTorch [37] with custom CUDA kernels from FlashInfer [59] and Triton [48].

## 6.1 Setup

**Models.** We test dense Llama-2 models [49], sparse mixture of experts Mixtral models [17], multimodal LLaVA image [27] and video models [62], and API model OpenAI's GPT-3.5. For open-weight models, the number of parameters ranges from 7 billion to 70 billion, and we use float16 precision.

**Hardware.** We run most experiments on AWS EC2 G5 instances, which are equipped with NVIDIA A10G GPUs (24GB). We run 7B models on a single A10G GPU and larger models on multiple A10G GPUs with tensor parallelism [44]. We run some additional experiments on A100G (80GB) GPUs.

**Baselines.** We compare SGLang against both high-level programming systems with their respective languages and default runtimes, as well as low-level inference engines with standard OpenAI-like Completion APIs. Unless otherwise stated, we do not turn on optimizations that will change the computation results so that all systems compute the same results. The baselines include:

- Guidance[13], a language for controlling LLMs. We use Guidance v0.1.8 with llama.cpp backend.
- vLLM [23], a high-throughput inference engine. We use vLLM v0.2.5 and its default API server[3].
- LMQL [4], a query language. We use LMQL v0.7.3 with Hugging Face Transformers backend.

**Workloads.** We test the following: 5-shot MMLU [14] and 20-shot HellaSwag [61] benchmarks. We decode one token for MMLU and use primitive `select` to select the answer with the highest probability for HellaSwag. For the ReAct agent [57] and generative agents [36], we extract the traces from the original papers and replay them. We use the Tree-of-thought [56] for the GSM-8K problems and Skeleton-of-thought [33] for tip generation. We use LLM judges with the branch-solve-merge [40] technique; JSON decoding with a schema specified by a regular expression; Multi-turn chat with 4 turns, where the input of each turn is randomly sampled between 256-512 tokens. Multi-turn chat (short) means short output (4-8 tokens) and multi-turn chat (long) means long output (256-512 tokens); DSPy retrieval-augmented generation (RAG) pipeline [20] in its official example.

**Metrics.** We report two performance metrics: throughput and latency. For throughput, we run a sufficiently large batch of program instances to compute the maximum throughput, comparing the number of program instances executed per second (programs per second, p/s). For latency, we execute a single program at a time without batching and report the average latency for multiple instances.

## 6.2 End-to-End Performance

**Results on open-weight models.** The latency and throughput results are shown in Fig. 5 and Fig. 6. SGLang improves throughput by up to $6.4\times$ and reduces latency by up to $3.7\times$. These improvements

---

[3]RadixAttention has been partially integrated as an optional experimental feature into the latest version of vLLM; therefore, we used an earlier version for comparison.

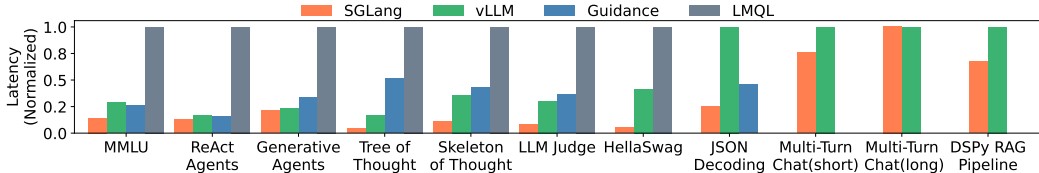

Figure 6: Normalized latency on Llama-7B models. Lower is better.

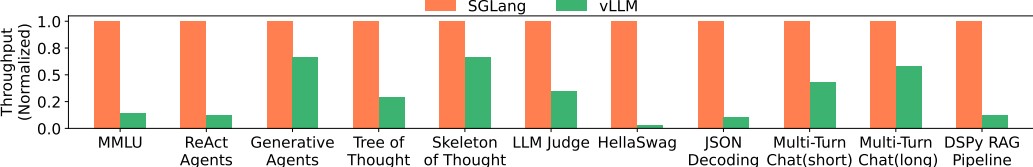

Figure 7: Normalized throughput on Mixtral-8x7B models with tensor parallelism. Higher is better.

result from KV cache reuse, the exploitation of parallelism within a single program, and faster constrained decoding. Next, we explain the reasons for the speedup in each benchmark.

On MMLU, SGLang can reuse the KV cache of the 5-shot examples with RadixAttention. RadixAttention benefits both throughput and latency. RadixAttention reduces total memory usage by sharing the KV cache, allowing for a larger batch size to improve maximum throughput. RadixAttention also reduces the computation of prefill, thus decreasing the first token latency. On HellaSwag, SGLang reuses the KV cache of both few-shot examples and the common question prefix for multiple choices, resulting in two-level sharing. For the ReAct and generative agents, SGLang reuses the KV cache of the agent template and previous calls. On Tree-of-thought and Skeleton-of-thought, SGLang parallelizes the generation calls within a single program and reuses the KV cache as much as possible. On JSON decoding, SGLang accelerates decoding by decoding multiple tokens at once with a compressed finite state machine. In multi-turn chat, SGLang reuses the KV cache of the chat history. The speedup is more noticeable for short outputs because KV cache reuse mostly helps reduce the prefix time. For long outputs, because there is not much sharing between different chat sessions and the decoding time dominates, there is almost no speedup. In the DSPy RAG pipeline, SGLang reuses the KV cache of the common context example. On these benchmarks, the cache hit rate ranges from 50% to 99%. Fig. 13 (Appendix) lists the achieved and optimal cache hit rates for all of them, showing that our cache-aware scheduling approaches 96% of the optimal hit rate on average.

We exclude LMQL and Guidance from some of the last five benchmarks due to slow performance and missing functionalities. LMQL's issues stem from slow token-level processing and an unoptimized backend, while Guidance lacks batching and parallelism support.

**Results on larger models with tensor parallelism.** We run larger models, Mixtral-8x7B and Llama-70B, with tensor parallelism on the same set of benchmarks and report the results in Fig. 7 and Fig. 12 (Appendix). The speedup on larger models shows a trend similar to that observed on smaller models, indicating that our optimization generalizes well to larger models. We omit Guidance and LMQL here because they lack efficient implementations of tensor parallelism.

**Results on multi-modal models.** SGLang has native support for multi-modal models with the `image` and `video` primitives. The optimizations in this paper are compatible with multi-modal models. For RadixAttention, we compute the hash of the input images and use it as the key in the radix tree, allowing us to reuse the KV cache of the image tokens from the same image. We run LLaVA-v1.5-7B (image) on llava-bench-in-the-wild and LLaVA-NeXT-34B (video) on ActivityNet. Because these models are not well supported by other baseline systems, we use the model authors' original implementation in Hugging Face Transformers as the baseline. As shown in Table 2, SGLang provides throughput up to $6\times$ higher on these benchmarks. In llava-bench-in-the-wild, there are multiple questions about the same image, and SGLang runtime reuses the KV cache in this case.

**Production deployment.** SGLang has been deployed in Chatbot Arena [8] to serve open-weight models. Due to low traffic for some models, only one SGLang worker serves each. After one month, we observed a 52.4% RadixAttention cache hit rate for LLaVA-Next-34B [28] and 74.1% for Vicuna-33B [7]. Cache hits come from common system messages, frequently reused example images, and multi-turn chat histories. This reduces first-token latency by an average of $1.7\times$ for Vicuna-33B.

Table 2: Throughput comparison on multi-modal LLaVA image and video models.

| Model | LLaVA-v1.5-7B (image) | LLaVA-NeXT-34B (video) |
|---|---|---|
| Author's original implementation | 0.18 image/s | 0.02 frame/s |
| SGLang | **1.15 image/s** | **0.10 frame/s** |

Figure 8: (a)(b) Cache hit rate ablation study. (c) RadixAttention ablation study.

**Results on API models.** We test a prompt that extracts three fields from a Wikipedia page using OpenAI's GPT-3.5 model. By using few-shot prompting, the accuracy of API speculative execution is high, and it reduces the cost of input tokens by about threefold due to the extraction of three fields.

## 6.3 Ablation Study

**Cache hit rate vs. latency/throughput.** Fig. 8(a)(b) shows the relationship between cache hit rate and performance metrics (first token latency, total latency, batch size, and throughput) on the tree-of-thought benchmark. The figure is obtained by partially disabling matched tokens at runtime. It shows that a higher cache hit rate leads to a larger batch size, higher throughput, and lower latency.

**Effectiveness of RadixAttention.** We test the effectiveness of RadixAttention and its components on several representative benchmarks. As shown in Fig. 8(c), "No Cache" means not using any cache, "No Tree-Structure" means using a simple table-based cache instead of a tree-structured cache, "FCFS Schedule" means using a first-come-first-serve policy instead of our cache-aware scheduling, "Random Schedule" means using a random order to schedule requests, "No Frontend Parallelism" means disabling parallelism in the interpreter, "No Frontend Hint" means disabling sending the fork hints from the interpreters, and "Full optimizations" means we turn on all optimizations. The experimental results show that each of these components is required to achieve the best performance. Disabling parallelism and hints from the frontend interpreter also results in suboptimal runtime performance, highlighting the importance of co-designing the frontend language and runtime.

**Overhead of RadixAttention.** We test the overhead of RadixAttention on a benchmark without any KV cache reuse opportunities. The benchmark measures throughput on the ShareGPT dataset. It takes 74.3 seconds to run 100 requests; however, the time used for managing the RadixAttention data structures is only 0.2 seconds, which is a negligible overhead of less than 0.3%. This is because the complexity of tree operations is linear and small. Thus, we can turn on RadixAttention by default.

**Effectiveness of the compressed finite state machine.** We test the effectiveness of the compressed finite state machine and its components on the JSON decoding benchmark. Experimental results show that the compressed finite state machine increases the throughput by $1.6\times$ because it can decode multiple tokens at once. In addition, we need to preprocess the state machine and reuse it for a batch of requests. Otherwise, redoing the preprocessing for each request makes the throughput $2.4\times$ lower.

## 7 Related Work

Various works have explored the reuse of the KV cache, and many of them are concurrent with our work. Uniquely, our RadixAttention first proposes treating the KV cache as a tree-based LRU cache. It is the first solution that supports multi-level sharing, cache-aware scheduling, frontend-runtime co-scheduling, and distributed cases. vLLM [23] and ChunkedAttention [58] explore some simple reuse cases (e.g., system prompt sharing) but do not cover multi-level tree-structured sharing or LRU caching. PromptCache [12] proposes the modular reuse of the KV cache beyond the prefix but can impact accuracy by up to a 43% drop. HydraGen [18], FlashInfer [59], and ChunkedAttention [58] focus on CUDA kernel optimizations and do not include the concept of an LRU cache. InferCept [1] and LLM-SQL [29] study KV cache reuse for specific applications such as interleaving with external API calls and relational databases, but they do not have our radix tree or cache-aware scheduling.

Several LLM programming and agent frameworks exist, such as Guidance [13], LMQL [4], DSPy [20], LangChain [24], AutoGen [55], and LLM Compiler [21]. Guidance and LMQL are most similar to SGLang, and we compare them in Sec. 2. Our innovation lies in novel runtime optimizations for accelerating the proposed programming model. SGLang is compatible with other frameworks and can accelerate them (e.g., the DSPy example in our evaluation). Additionally, SGLang is compatible with many other common inference optimizations [60, 39, 3, 23, 59, 10, 26, 15, 19, 32, 31, 11].

# 8 Future Directions and Conclusion

**Future directions.** Despite the progress made with SGLang, several limitations remain that reveal promising directions for future research. These include extending SGLang to support additional output modalities, adapting RadixAttention to operate across multiple levels of the memory hierarchy (e.g., DRAM, Disk) [43], enabling fuzzy semantic matching within RadixAttention, providing higher-level primitives atop SGLang, fixing starvation in cache-aware scheduling [42], and enhancing the SGLang compiler to perform advanced static optimizations such as scheduling and memory planning.

**Conclusion.** We introduce SGLang, a framework for efficient programming and executing structured language model programs. SGLang significantly improves the throughput and latency of complex LM programs through novel optimizations like RadixAttention, compressed finite state machines, and a language interpreter. It is a valuable tool for developing advanced prompting techniques and agent workflows. The source code is publicly available.

# Acknowledgement

This project is supported by the Stanford Center for Automated Reasoning and gifts from Astronomer, Google, IBM, Intel, Lacework, Microsoft, Mohamed Bin Zayed University of Artificial Intelligence, Nexla, Samsung SDS, Uber, and VMware. Lianmin Zheng is supported by a Meta Ph.D. Fellowship. We thank Yuanhan Zhang and Bo Li for the LLaVA-NeXT (video) support.

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

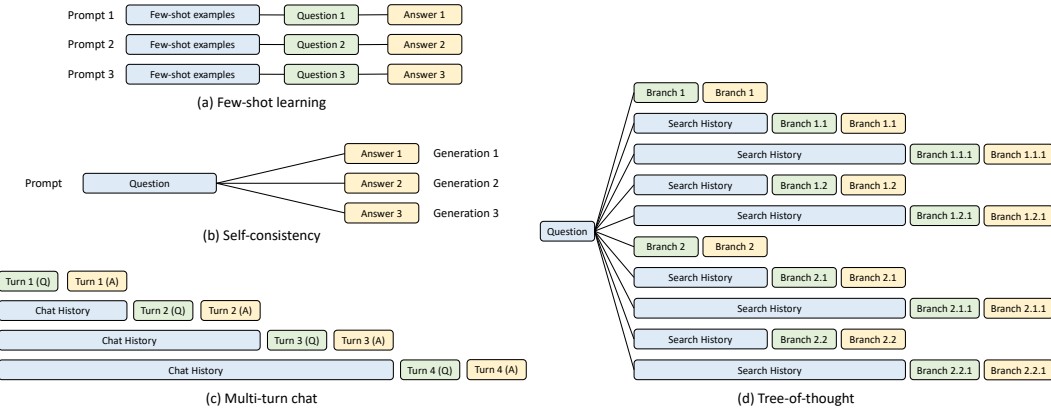

Figure 9: KV cache sharing examples. Blue boxes represent shareable prompt parts, green boxes indicate non-shareable parts and yellow boxes mark non-shareable model outputs. Shareable elements include few-shot learning examples, questions in self-consistency [53], chat history in multi-turn chat, and search history in tree-of-thought [56].

# A  Additional Details on RadixAttention

## A.1  Background on the KV Cache

Most LLMs in use today, such as GPT-3 [5], PaLM [9], and LLaMA [49], are based on the autoregressive Transformer architecture [51]. These models predict the probability of the next token in a sequence based on the preceding tokens. During inference, the model first processes a sequence of input tokens through a forward pass (this process is called "prefill"). It then sequentially decodes output tokens, with each token depending on prior tokens (this process is called "decoding"). We refer to the process of taking a sequence of input tokens and generating a sequence of output tokens as a single-generation call. Throughout this process, each token generates some intermediate tensors, which are used for decoding further tokens. These intermediate tensors, known as the "KV Cache," are named for the key-value pairs in the self-attention mechanism. An important observation when discussing optimizations in this paper is that the computation of the KV cache only depends on all previous tokens, so different sequences with the same prefix can reuse the KV cache of the prefix tokens and avoid redundant computation.

Often in LLM programs, multiple text segments and generation calls are appended to a single prompt. Caching the computed KV cache for previous tokens across multiple chained calls can reduce redundant computation. This optimization, however, is neither free nor trivial, as it requires additional storage and more complex memory management. In addition, it is common in LLM programs to generate multiple outputs from a single prompt or to fork a new prompt from the current state [25]. Basic prefix sharing has been investigated in vLLM [23]. More advanced sharing patterns like irregular tree-structured sharing can also be employed. Fig. 9 shows four typical patterns of KV cache sharing across multiple calls; none of the existing systems can automatically handle all of them. On the contrary, our RadixAttention in Sec. 3 can handle all of them automatically at runtime.

## A.2  Pseudocode for Cache-Aware Scheduling

Alg. 1 shows the pseudocode of cache-aware scheduling for RadixAttention with continuous batching.

## A.3  Proof of the Theorem 3.1

**Theorem 3.1.** *For a batch of requests, we can achieve an optimal cache hit rate by visiting the radix tree of the requests in the depth-first search order, with a cache size ≥ the maximum request length. The longest-shared-prefix-first order is equivalent to a depth-first search order.*

*Proof.* First, we show that the depth-first search (DFS) order achieves an optimal cache hit rate. Let $R$ denote the set of requests in the batch, and $T$ denote the radix tree built from $R$. For each edge $e$ of $T$, the KV cache associated with $e$ needs to be computed at least once. Let $|e|$ denote the size of the KV cache associated with $e$. Let $C$ denote the computational complexity of the KV cache for $R$.

---
**Algorithm 1** Cache-Aware Scheduling for RadixAttention with Continuous Batching.
---
**Input:** The radix tree $T$, the memory pool $P$, the current running batch $B$, the waiting queue $Q$.
**Output:** Finished requests and updated system state.
  // Get all requests from the waiting queue
  $requests \leftarrow Q$.get_all_requests()
  // Search for the prefix matching for all waiting requests
  **for** $req \in requests$ **do**
    $req.prefix\_node, req.prefix\_len \leftarrow T$.match_prefix($req.input\_tokens$)
  **end for**
  // Sort the requests according to the matched prefix lenghts
  $requests$.sort()
  // Select requests for the next batch
  $available\_size \leftarrow T$.evictable_size() $+ P$.available_size()
  $current\_size \leftarrow 0$
  $new\_batch \leftarrow []$
  **for** $req \in requests$ **do**
    **if** $req$.size() $+ current\_size < available\_size$ **then**
      $new\_batch$.append($req$)
      $delta \leftarrow T$.increase_ref_counter($req.prefix\_node$)
      $current\_size \leftarrow current\_size + delta$
    **end if**
  **end for**
  $Q$.remove_requests($new\_batch$)
  // Insert requests into the current running batch
  $B$.merge($new\_batch$)
  // Allocate new memory and do eviction if necessary
  $needed\_size \leftarrow B$.needed_size()
  $success, buffer \leftarrow P$.alloc($needed\_size$)
  **if** not $success$ **then**
    $T$.evict($needed\_size$)
    $success, buffer \leftarrow P$.alloc($needed\_size$)
  **end if**
  $B$.run($buffer$)
  // Process finished requests
  $finished\_requests \leftarrow B$.drop_finished_requests()
  **for** $req \in finished\_requests$ **do**
    $T$.decrease_ref_counter($req.prefix\_node$)
    $T$.insert($req$)
  **end for**
  **return** $finished\_requests$
---

We obtain the lower bound

$$C \geq \sum_{e \in \text{edges}(T)} |e|.$$

Consider we visit the radix tree $T$ in DFS order. For each edge $e$ of $T$, the first time we compute the KV cache associated with $e$, we will then compute the whole subtree of $e$. During the computation of the subtree of $e$, the edge $e$ will be continuously hit, so no additional computation will happen. After finishing the computation for the subtree rooted at $e$, the edge $e$ will not be visited again. Notice that, with a cache size $\geq$ the maximum request length, which equals the longest path in the radix tree $T$, edge $e$ will not be evicted during the computation of its subtree, since the common prefix including $e$ of the subtree will be continuously hit. Therefore, the KV cache associated with each edge $e$ will be computed only once. Thus, we achieve the lower bound

$$C = \sum_{e \in \text{edges}(T)} |e|.$$

The cache hit rate, defined as

$$\frac{\sum_{r \in R} \text{number of cached prefill tokens in } r}{\sum_{r \in R} \text{number of prefill tokens in } r},$$

equals $1 - \frac{C}{\sum_{r \in R} \text{number of prefill tokens}}$, reaches its upper bound, delivering optimality.

Next, we show that the longest-shared-prefix-first order is equivalent to a DFS order by induction.

- (Base) In the beginning, since there is nothing cached, a random request that corresponds to a node $x$ in $T$ will be processed. All requests that correspond to the nodes $\{v_1, ..., v_n\}$ on the path from the root to $x$ do not need a recomputation. The computation complexity for requests corresponding to the nodes $\{v_1, ..., v_n, x\}$ is aligned with a valid DFS. The path from the root to $x$ is cached.
- (Induction) Assume we just visited a node $y$ in $T$, and the visited nodes align with a DFS order. Let $P$ denote the path from the root to $y$. Then each node that has not been visited has the lowest common ancestor with the visited nodes on $P$. Since nodes on $P$ are cached, a node $z$ that has not been visited with the lowest common ancestor on $P$ will have the longest shared prefix. The longest-shared-prefix-first order will select $z$, which is a valid DFS order. The path from the root to $z$ will be cached because it is the most recent.

□

In the online case, the DFS order will be disrupted, but the longest shared prefix schedule still approximates the DFS behavior on the augmented part of the implied radix tree. We show this by considering the step of adding a new batch of requests.

Let $T$ denote the part of the radix tree that has been visited so far, and $T'$ denote the whole new radix tree after adding the new batch of requests. Let $C$ denote the set of cached nodes in $T$. Let longest($C$) denote the node in $C$ that has the longest path from the root and has a subtree in $T'$ that has not been fully visited.

The longest shared prefix schedule will then process the subtree in $T'$ rooted at longest($C$) in a DFS order. During this process, eviction could occur, and the remaining cached nodes from $C$ become $C^{(1)} \subseteq C$. A DFS will then occur for the subtree in $T'$ rooted at longest($C^{(1)}$).

Similarly, we will have $C^{(2)}, \ldots, C^{(k)}$, until $C^{(k)}$ contains only one leaf node in $C^{(k)}$ that has its subtree in $T'$ not fully visited. At this point, we have reached a valid DFS state. The remaining part of $T'$ will be visited in DFS order as described in the proof of Theorem 3.1.

### A.4 Data-Parallel Distributed RadixAttention

To adapt the RadixAttention for distributed settings with multiple replica workers (i.e., data parallelism), we developed a mechanism wherein each worker maintains its own sub-tree, while the router oversees a meta-tree. This meta-tree acts as a trie that tracks all sub-trees and their associated devices. Upon the arrival of a new batch of requests at the router, prefix matching is executed on the meta-tree. We implement various policies based on each request's affinity—measured by the length of the shared prefix with specific workers and other requests in the same group—to make efficient dispatch decisions that minimize redundant computations. Each time new requests are processed, both the router and workers independently update their respective trees. Should an eviction occur at a worker node, it commits this eviction to a queue, which the router then processes to update the meta-tree during periods of low activity. We benchmarked this distributed configuration using four workers and the MMLU dataset, observing that it achieves linear scaling and an optimal cache hit rate with minimal overhead from this weakly consistent distributed cache design. There exists a trade-off between maximizing data locality and parallel processing efficiency. Exploring advanced scheduling policies to optimize this trade-off is designated as an area for future research. In addition, concurrent work from Preble [45] studies data-parallel scheduling based on an early version of SGLang.

## B  Additional Details on Compressed Finite State Machine

This section discusses the background and implementation details of the compressed finite state machine for faster constrained decoding. We aim for LLMs to follow a regular expression (regex),

which offers greater expressiveness and can be used to represent common formats such as JSON schemas. To achieve this, we convert the regex into a Finite State Machine (FSM) to guide the generation process during decoding [54]. An FSM is essentially a graph with nodes (states) and edges (transitions with strings/characters). Starting from an initial state, each transition appends the string on the edge to move to the next state, using a set of final states to conclude the process. This mechanism guides an LLM's decoding, filtering out invalid tokens based on the FSM's current state transitions, as illustrated in  Fig. 10. The decoding might involve multiple transitions in the FSM until reaching a final state.

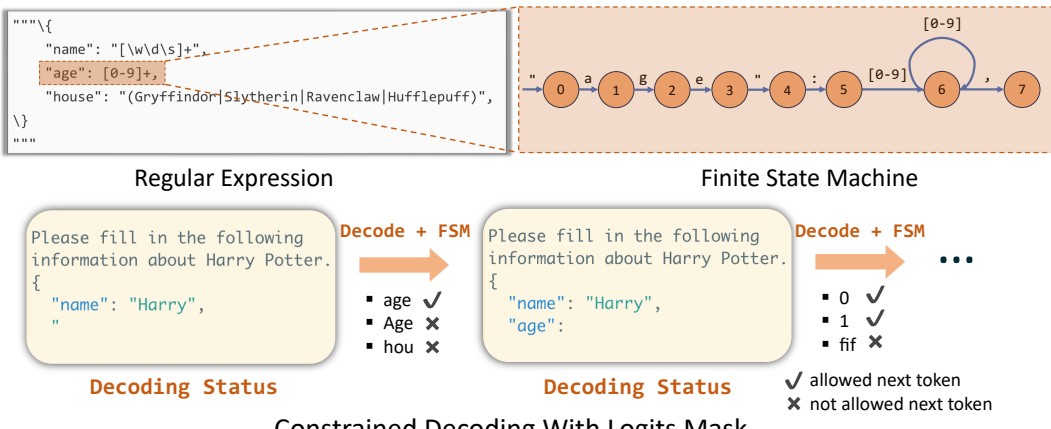

Figure 10: Example of how regex is converted into FSM and how FSM guides the decoding process.

The challenge of constrained decoding arises from the fact that constraints are often expressed in natural language formats, i.e., regex is depicted by characters/strings, while LLMs are designed to interpret and process these as tokens. This creates a complex scenario since the mapping between strings and tokens is intricate and lacks a one-to-one correspondence  [22].

This section is derived from an earlier blog post[4]. Readers are also encouraged to read the blog post for additional background and easier understanding.

### B.1  Implementation Details of Compressed Finite State Machine

To simplify the construction of Compressed FSM, we build the original FSM on characters/strings instead of on tokens. We formally define the concepts of singular transition edge and compressed edge as follows:

- Singular transition edge: A edge is a singular transition edge if 1) its source node has only one successor node, and 2) there is only one acceptable character/string on it.
- Compressed edge: An edge compressed by several consecutively adjacent edges $(e_0, e_1, \ldots, e_k)$ if and only if $e_1, \ldots, e_k$ are singular transition edges. The text of the compressed edge is the concatenation of the texts of $e_0, e_1, \ldots, e_k$.

Starting from an FSM based on characters, we recursively merge singular transition edges into their preceding ones until further compression is unfeasible, resulting in a Compressed FSM. This approach speeds up the decoding process, as demonstrated by SGLang's runtime efficiency with the Compressed FSM, shown in  Fig. 11.

### B.2  Handling Tokenization Artifacts with Retokenization

When a new token is generated, we get the token's string and search all the outgoing edges of the current state find the one that starts with the just decoded string, and then move forward. However, when the transition edge is a well-compressed one and contains a very long string, we may anticipate the next rounds' decoded strings as well. This is where the acceleration happens and we call this process *Jump Forward*. However, we still need to convert the string into tokens for the next decoding phases and it is not straightforward due to the LLM's specific pretraining and tokenization method;

---

[4]https://lmsys.org/blog/2024-02-05-compressed-fsm/

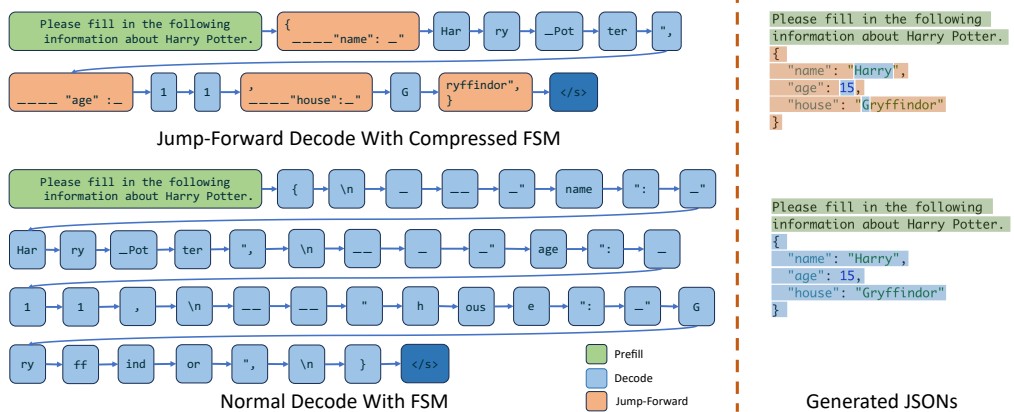

Figure 11: Comparison of decoding using Compressed FSM versus normal FSM: The left subfigure depicts the decoding process per forward pass, while the right subfigure explains the origins of various result components.

direct partitioning might alter the intended meaning [50]. For example, the compressed text in Fig. 2's regex is {"summary": ", which can only be tokenized as {", summary, ": and _" according to the tokenizer instead of partition them randomly such as {", summa, ry, and ":_". To address this issue, we use the original tokenizer to retokenize all the previous text as well as the text of the compressed edge, ensuring alignment with the original input format of LLMs. And it only brings minor retokenization overhead.

### B.3 Future Extension: Addressing Distorted Probability

The challenge caused by the gap between strings and tokens also brings the problem of skewed probability distribution [50]. For example, in Fig. 2, the regex "[ABCD][+-]?" suggests grades from A+ to D-, but if replaced with broader terms like Excellent|Above Average|Fair|Below Average, the runtime may inaccurately map an A to Above Average due to the term Above Average is on a compressed transition, misrepresenting the grade hierarchy. This occurs because the LLM doesn't recognize the specific range of choices, leading to inappropriate token sequences. Computing accurate probabilities for each choice requires summing the probabilities of all token sequences that result in each choice, which complicates decoding and adds overhead. One workaround is to include the choices or the regex directly in the prefill prompt, guiding the LLM to be aware of its choices and output the decision in proper token sequences. However, this approach doesn't solve the underlying issue of distorted probabilities, highlighting the need for further research to improve the compressed FSM's accuracy.

## C  Additional Experimental Setups and Results

**Additional experimental setups.** Fig. 5 and Fig. 6 are obtained by running Llama-7B on a single A10G (24GB) GPU. Fig. 7 are obtained by running Mixtral-8x7B on 8 A10G (24GB) GPUs with tensor parallelism. Fig. 8(c) is obtained by running Llama-7B on a single A10G (24GB) GPU. Fig. 12 are obtained by running Llama-70B on 4 A100G (80GB) GPUs with tensor parallelism. Table 2 are obtained by running LLaVA-v1.5-7B on a single A10G (24GB) GPU and running LLaVA-Next-34B on a single A100G (80GB) GPU. Each bar in the benchmark figures takes several minutes to an hour to run.

**Additional experimental results.** Fig. 13 shows the achieved and optimal cache hit rates on the benchmarks listed in Fig. 5. Fig. 12 shows the throughput on Llama-2-70B with tensor parallelism.

## D  Compiler Mode

Besides the interpreter mode used in the main body of the paper, another way to run SGLang programs is to compile them as computational graphs and execute them with a graph executor. This opens up

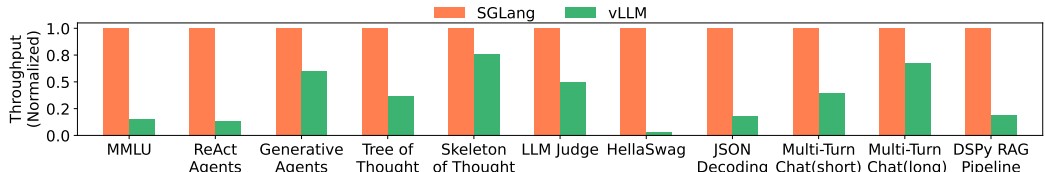

Figure 12: Normalized throughput on Llama-2-70B models with tensor parallelism. Higher is better.

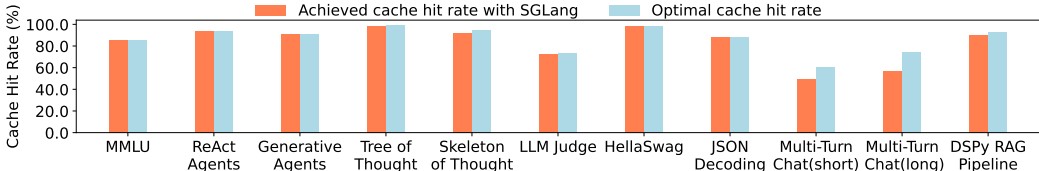

Figure 13: Achieved cache hit rate and optimal cache hit rate on various benchmarks.

opportunities for more compilation optimizations, as we can rewrite the graph and perform more static planning.

### D.1 Design and Implementation

We designed an intermediate representation (IR) for SGLang, which represents SGLang program structures and operations as a computational graph. This graph includes nodes for primitive operators and edges for dependencies. See Fig. 14b for the graph corresponding to the program in Fig. 14a. In the program, each call to a decorated function or fork creates a new prompt state or a stream.

There are several types of nodes. Each operand of the operators += and + in a SGLang program is represented by an IR node. These include `ConstantText`, `Argument`, `Gen`, `Select`, `Variable`, `Fork`, `GetForkItem`, and `Join`. There are two types of dependencies: intra-stream dependency, where operations submitted into a stream using += must be executed after all preceding operations in that stream, and inter-stream dependency, which occurs when one stream needs to fetch variable values from another stream, necessitating synchronization. Operations like fork manipulate multiple streams and thus introduce inter-stream dependencies.

To generate the graph, we use tracing to run the program with abstract arguments and construct the graph dynamically. This method is limited to programs without data-dependent control flow, a limitation we plan to address in future work. Once constructed, the graph can be executed through a graph executor, eliminating the need to reinterpret the original Python program. This results in benefits like graph rewriting optimizations, reduced runtime overhead, and program serialization. For execution, stream executors are launched for each data stream, dispatching IR nodes to the streams in topological order.

### D.2 A Case Study of Compiler Optimization: Code Movement for Improving Prefix Sharing

We explore a compilation optimization for SGLang IR: code movement for improving prefix sharing. We anticipate that more classical compiler techniques can also be applied, such as auto-tuning and instruction selection.

This optimization aims to improve prefix sharing by reordering nodes in the graph to increase the length of the constant prefix. It does not strictly preserve the original computation, classifying it as an aggressive optimization. For instance, changing the prompt from "Here is a question + {question}. Please act as a math expert and solve the given question." to "Please act as a math expert and solve the given question. Here is a question + {question}." results in a longer shareable prefix. This optimization is interesting because traditional program analysis cannot achieve it due to the presence of natural language instructions in SGLang. Instead, we prompt GPT-4 to reorder graph nodes. We write a prompt with several examples to teach GPT-4 the concepts of SGLang IR, and we find that GPT-4 can successfully apply this optimization for some simple SGLang programs.

```
@function
def expand(s, tip):
    s += (
        "Please expand the following tip into a "
        "detailed paragraph: " + tip + "\n"
    )
    s += gen("paragraph")

@function
def tip_suggestion(s, topic):
    s += "Here are 2 concise tips for " + topic + ".\n"

    # Generate a skeleton of two short tips
    s += "1." + gen("tip_1", stop=["\n", ":", "."]) + "\n"
    s += "2." + gen("tip_2", stop=["\n", ":", "."]) + "\n"

    # Expand the tips into detailed paragraphs in parallel
    detailed_tip1 = expand(tip=s["tip_1"])
    detailed_tip2 = expand(tip=s["tip_2"])

    # Merge the results and generate a summary
    s += "Tip 1: " + detailed_tip1["paragraph"] + "\n"
    s += "Tip 2: " + detailed_tip2["paragraph"] + "\n"
    s += "In summary" + gen("summary")

state = tip_suggestion.run(topic="staying healthy")
print(state.text())
```

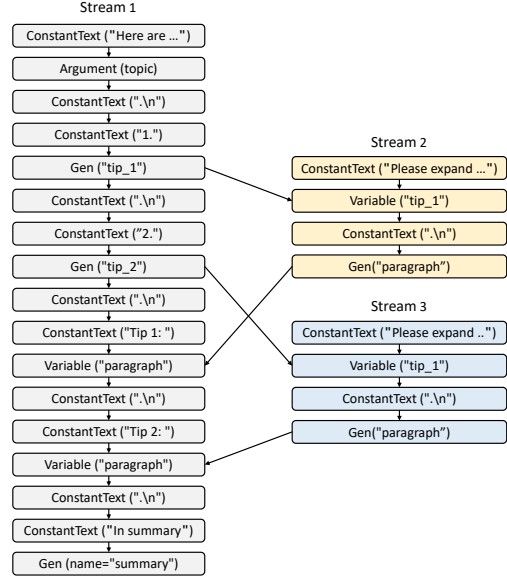

(a) The SGLang program for parallel tip suggestion with skeleton-of-thought prompting.

(b) A computational graph for the program in Fig. 14a. The three streams correspond to three function calls.

Figure 14: An SGLang program and its corresponding dataflow graph.

We evaluate the effectiveness of this optimization. We collect 20 prompt templates from the internet and implement them in SGLang. We utilize 5 of these templates as few-shot training examples and the remaining 15 as test cases. Our evaluation shows that, for 12 out of the 15 templates, GPT-4 successfully reorders the graph nodes without altering the semantics, as confirmed by manual inspection of the modified prompts. On average, this optimization results in a 60-token increase in the shareable prefix length, showcasing GPT-4's effectiveness. Failures in creating optimized prompt order come from an incorrect understanding of the semantic meaning behind the graph nodes. It is too aggressive and puts all constants upfront even when such ordering changes the original semantics. This case study aims to explore the use of GPT-4 for compiler optimizations. More work is needed to make these kinds of optimizations reliable in the future.

