# OpenReview forum: "SGLang: Efficient Execution of Structured Language Model Programs"
_NeurIPS.cc/2024/Conference — NeurIPS 2024 poster_

### Official Review · Reviewer_N1WL · 2024-07-12

**Soundness:** 3
**Presentation:** 2
**Contribution:** 4
**Rating:** 7
**Confidence:** 4

**Summary:**

This paper proposes SGLang, a system with  frontend programming interface and run-time optimization for efficient execution of LLM-based applications. For frontend programming interface, SGLang provides primitives for generation and parallelism control. For run-time optimization, SGLang has three key optimizations. The first is to maintain a LRU cache of the KV cache for all generation requests within a radix tree to enable the automatic reuse of KV cache across requests. The second is to build a compressed finite-state machine (FSM) to represent the constraint of the request with structured outputs. Through analyzing the  FSM, SGLang is able to decode multiple tokens of the given structured template at a time. The last is to reduce the number of endpoint API call with a speculative decoding mechanism on the first API call. Experimental results show that SGLang achieves up to 6.4× higher throughput compared to state-of-the-art inference system.

**Strengths:**

1. This paper focuses on solving a critical problem: efficient inference of LLM-based applications.
2. Instead of designing a new algorithm for efficient execution of LLM-basd application, this paper optimize from the prompt aspects, which is convenient to be applied and causes no harm for the accuracy.
3. The author built a comprehensive system, containing both frontend programming interface and run-time optimizations and covering different optimization strategies.
4. Experimental results shows that SGLang achieves a significant higher throughput compared to state-of-the-art inference system.

**Weaknesses:**

1. **No modular sensitivity study:** SGLang incorporates optimizations from various perspectives. These strategies are independent and can be applied individually or in combination. However, their effectiveness may vary depending on the context. Therefore, I recommend to include a sensitivity analysis in experiments to evaluate the specific benefits of each optimization strategy in enhancing inference efficiency.

2. **Limited real-world applicability:** The practical applicability of FSM and API speculative decoding is questionable in real-world scenarios. These optimizations appear to enhance efficiency primarily in specific instances, such as when processing JSON outputs with predefined structures or when executing multiple API calls within a single sentence.

3. **Add more baselines for RadixAttention evaluation:** In related works, the author noted that many existing studies have devised their own approaches for reusing key-value (KV) caches, emphasizing the distinctive nature of the designed RadixAttention. To further demonstrate the effectiveness of RadixAttention, it is recommended to include additional baselines such as PromptCache and API Serve in the evaluation.

**Questions:**

See above.

**Limitations:**

N.A.

---

> ### Author Rebuttal · Authors · 2024-08-07
>
> We appreciate your valuable feedback. We will incorporate the clarifications and address the issues in the next draft. Here is our response to your questions:
>
>
> > Weaknesses 1: No modular sensitivity study.
>
> Section 6.3, "Ablation Study," in the original paper did what you asked for. In line 306, we conducted the sensitivity study of our first technique, RadixAttention. In line 321, we demonstrated the effectiveness of our second technique, the compressed finite state machine. In line 298 (section 6.2), we showed the effectiveness of our third technique, API speculative execution.
>
> All techniques proposed in this paper include standalone sensitivity analyses. We will highlight these results more in the next version.
>
> > Weaknesses 2: The practical applicability of FSM and API speculative decoding is questionable in real-world scenarios.
>
> Both optimizations originate from the real workloads we observed in production.
>
> The compressed FSM for constrained decoding is a very general technique that can accelerate almost all constrained decoding workloads. OpenAI just released [a new feature on structured decoding](https://openai.com/index/introducing-structured-outputs-in-the-api/) this week, as this is one of the most desired features by the developers.  We tested the first two official examples and found that the compressed FSM can achieve a 1.3x to 1.6x speedup.
>
> The API speculative decoding originates from some workloads from DSPy, a retrieval-argument-generation workflow. It typically uses few-shot examples to enforce a format and requires a single call to generate multiple fields. API speculative decoding can greatly reduce the cost of this.
>
>
> > Weakness 3: Add more baselines for RadixAttention evaluation
>
> Thanks for the suggestion. We compared SGLang with the open-source code of PromptCache.
>
> In terms of the idea, PromptCache focuses more on the modular reuse of the KV cache to reduce time-to-first-token latency. SGLang, on the other hand, is about the tree-structured automatic KV cache reuse. They did something similar, but SGLang provides a more general, automatic, and lossless solution.
>
> Although their original focuses are slightly different, we can still run the two systems on the same workload and compare their performance. We used their default recommended API to run the same workload, with both systems' caches turned on. Here are the results:
>
> | Framework                        | Time-to-first-token (ms, lower is better) | Decoding throughput (tokens/s, higher is better) |
> |----------------------------------|------------------------------------------|--------------------------------------------------|
> | PromptCache (batch size = 1) | 180.2                                     | 9.2                                              |
> | SGLang (batch size = 1)      | 21.3                                      | 80.4                                             |
> | SGLang (batch size = 64)     | 729.3                                     | 1302.8                                           |
>
> We found that SGLang provides two major advantages:
> 1. SGLang has much better system optimizations. With a batch size of 1, the time-to-first-token latency of SGLang is 8.5x better than PromptCache. This is because SGLang uses highly optimized kernel implementation and has a much better runtime implementation. The decoding speed of SGLang is also about 8.7x faster.
> 2. SGLang supports continuous batching and can greatly improve the decoding throughput with a larger batch size.
>
> ---
>
>
> Please let us know if the answer addresses the weakness. We would greatly appreciate it if you could raise your score if you find the answers helpful.

---

> > ### Comment · Reviewer_N1WL · 2024-08-10
> > **Response**
> >
> > Thank the authors for the detailed response and clarification. The rebuttal has addressed my concern and I will raise my score.

---

### Official Review · Reviewer_U8WY · 2024-07-12

**Soundness:** 4
**Presentation:** 4
**Contribution:** 4
**Rating:** 7
**Confidence:** 4

**Summary:**

This paper introduces SGLang, a programming language for large language models (LLMs) that enables users to write programs specifying prompts in natural language and control flow with Python and execute them to call LLMs as needed. ​ SGLang provides a set of language constructs and functions that allow users to express complex tasks especially parallelizing multiple LLM calls. ​ The paper presents various optimizations and techniques for improving the efficiency and performance of LLMs, such as reusing  KV cache across multiple LLM calls and compressed finite state machines to decode multiple tokens simultaneously during constrained decoding. ​ Experimental results demonstrate the effectiveness of these techniques in improving the throughput of various LLMs on various tasks. ​

**Strengths:**

The paper is well written. I thoroughly enjoyed reading the paper. The authors have done extensive exploration of the system space. Every alternative of the system that a reader could have think of, they have tried to answer it. For example, the authors present both an interpreter version and  a compiler version to execute SGLang programs; they consider multiple modes of accessing the models such as batch mode and sequential mode. In-addition to implementing cache reuse, they also explored cache-aware scheduling algorithm to further take advantage of the cache. They even handle multiple models sizes including those requiring distribution across multiple GPUs.

The experiments are also very extensive and try to assess the value of each component individually and together and show a clear empirical advantage of this system over other existing system.

Overall, the paper is interesting and well thought-out, showcasing significant contributions to the field.

**Weaknesses:**

While the paper covers many interesting aspects, some parts did not receive as much attention. For example, optimizing for API-based model calls is not thoroughly explained or explored, but that is also not the main goal of the paper. Hopefully, future papers will handle and explore them more thoroughly.

Minor: consider using phrases like "improves throughput by x times to y times" instead of  phrases like "improves throughput up to 6.4x"

**Questions:**

1. How does the model handle model failures especially with token limitations and timeouts with API models? Does the approach give control to users to specify how they would want to handle these failures?

2. The experiment report on the effectiveness of the compressed finite state machine is not very clear. Can you explain why compressing the state machine can lead to lower throughputs in some cases and under what circumstances this occurs?

**Limitations:**

Limitations are addressed.

---

> ### Author Rebuttal · Authors · 2024-08-07
>
> We appreciate your valuable feedback. We will incorporate the clarifications and address the issues in the next draft. Here is our response to your questions:
>
> >  For example, optimizing for API-based model calls is not thoroughly explained or explored, but that is also not the main goal of the paper.
>
> We will add a detailed example in the main body if space allows, and in the appendix otherwise.
>
> > How does the model handle model failures especially with token limitations and timeouts with API models? Does the approach give control to users to specify how they would want to handle these failures?
>
> Yes, SGLang has error handling APIs as well. They are not introduced in the paper due to space limitations. Basically, the state will record any errors it encounters. After an SGLang program finishes or terminates due to errors, the user can fetch all error details. Depending on the error type, users can write their code to handle or retry them.
>
> > Can you explain why compressing the state machine can lead to lower throughputs in some cases and under what circumstances this occurs?
>
> Sorry for the confusion. Line 325 reads: "Otherwise, redoing the preprocessing for each request makes the throughput 2.4× lower." This means that without our cached preprocessing optimizations, redoing the preprocessing every time will be much slower. Please note the word "Otherwise" at the beginning. This emphasizes that our optimization is very effective. We will improve the wording in the next version.
>
> ---
>
> Please let us know if the answer addresses the weakness. We would greatly appreciate it if you could raise your score if you find the answers helpful.

---

> > ### Comment · Reviewer_U8WY · 2024-08-13
> > **Response**
> >
> > Thank you for your response; that clarifies my questions.

---

### Official Review · Reviewer_PeTj · 2024-07-12

**Soundness:** 4
**Presentation:** 4
**Contribution:** 4
**Rating:** 8
**Confidence:** 3

**Summary:**

The paper introduces SGLang, a system designed for efficient execution of complex language model (LM) programs. SGLang consists of a frontend language that simplifies programming with generation and parallelism primitives, and a runtime that boosts execution speed through several optimizations. The paper describes two main contributions: RadixAttention for Key-Value cache reuse, and a compressed finite state machine for constrained decoding. It also describes a method for API speculative execution. The evaluation demonstrates that SGLang achieves significantly higher throughput and lower latency on various tasks when using open-weights models.

**Strengths:**

* Quality: The evaluation is thorough, with experiments conducted on a diverse set of LLM workloads, including agent control, logical reasoning, few-shot learning benchmarks, JSON decoding, retrieval-augmented generation pipelines, and multi-turn chat. The results show substantial improvements in throughput and latency for open-weights models. I would also like to commend the authors for clear and transparent description of the challenges and limitations of the work.
* Clarity: The paper is very well written, providing a clear overview of the problem, detailed explanations of the contributions, and comprehensive descriptions of the evaluation methodology. I like the figures, which I thought are helpful in understanding the overall system and the contributions of the paper.
* Significance: In my opinion, this work is highly relevant to current developments in programming with large language models. The tools and techniques introduced in SGLang are likely to be valuable for practitioners working on complex LM applications.
* Originality: I am not familiar well enough with the related work to be able to evaluate the originality of the paper.

**Weaknesses:**

If I have one weakness to mention, it would be that the contribution and evaluation related to API models seem relatively minor compared to the other contributions. Providing more details on optimizing API calls to public API models and including more examples in the evaluation would strengthen this aspect of the paper (in addition to being valuable to practicioners).

**Questions:**

Have you tried any other examples for the public API evaluation, or is it only the wikipedia extraction example? In general, what class of LM program is well-suited for those optimisations?

Minor comments and suggestions for improvements:
*  line 48: is there a missing “at” in “multiple tokens can often be decoded (at) once”?

**Limitations:**

I thought the limitations were well and clearly addressed and I liked the wide range of suggestions for future work.

---

> ### Author Rebuttal · Authors · 2024-08-07
>
> We appreciate your valuable feedback. We will incorporate the clarifications and address the issues in the next draft. Here is our response to your questions:
>
> > Providing more details on optimizing API calls to public API models and including more examples in the evaluation would strengthen this aspect of the paper.
>
> We will add a detailed example in the main body if space allows, and in the appendix otherwise.
>
> > Have you tried any other examples for the public API evaluation, or is it only the wikipedia extraction example? In general, what class of LM program is well-suited for those optimisations?
>
> We also tested some DSPy retrieval-augmented generation pipelines. This optimization is inspired by these real-world workloads. In general, an LM program with few-shot examples that asks for multiple fields in the output will be well-suited for these optimizations. The few-shot examples can hint at the format, thereby increasing the success rate of speculative execution. Asking for multiple output fields is also necessary to make this optimization effective because it can automatically merge them into a single call.

---

> > ### Comment · Reviewer_PeTj · 2024-08-10
> > **Response**
> >
> > Thank you for your response. I'm looking forward to reading the next version of this paper when available.

---

### Official Review · Reviewer_P67b · 2024-07-15

**Soundness:** 4
**Presentation:** 4
**Contribution:** 3
**Rating:** 7
**Confidence:** 3

**Summary:**

The authors introduce and evaluate SGLang, a LLM language and implementation that can perform inference or use an external API. They claim significant performance improvements mostly though cache redesign.

**Strengths:**

I liked the Python-embedded language. It seems relatively straightforward to use. The results look also pretty good for the cache, less so for the reg exps. The authors also claim good results in interfacing with Chat-gpt3.

**Weaknesses:**

The authors provide a  nice example of their language, but very informal. Is the language just what you mention here.?

Radix trees are a good solution for data that tend to have the same prefix. This is common in web-data, such as URIs, but may not be so worthwhile in other types of data. It would be helpful to see how much this technique across the different benchmarks.

How do you select the least recently used cache entry?

I wonder whether your scheduling strategy may lead to starvation.

Section 4 is mostly a pointer  to App B,

I missed a description of the absolute times. One thing is to speedup 1 ms. and another 1000 days, Also, I imagine this will take as much GPU memory as it can take, it would be nice to evaluate the cost of  memory usage,

Finally you mention speeding for paralleillsim. One other important consideration is cost. YOu seem to do that, but I could  not find that clearly in the paper.

**Questions:**

My questions were asked above.

**Limitations:**

The system above may have a significant impact on the community, hopefully  positive.



The authors do focus on speeding existing LLMs, only.

---

> ### Author Rebuttal · Authors · 2024-08-07
>
> We appreciate your valuable feedback. We will incorporate the clarifications and address the issues in the next draft. Here is our response to your questions:
>
>
> > The authors provide a nice example of their language, but very informal. Is the language just what you mention here.?
>
> We introduced the language informally, covering 90% of its features. While we refer to it as a "language," it's similar to how NumPy and PyTorch are considered "array/tensor computation languages embedded in Python." If needed, we can provide a more formal description.
>
> The language is designed to be intuitive and extensible, allowing you to easily add new primitives. Our submitted code includes more APIs and usage examples. Many researchers have adopted this language and used it to develop various workflows.
>
> > Radix trees are a good solution for data that tend to have the same prefix. This is common in web-data, such as URIs, but may not be so worthwhile in other types of data. It would be helpful to see how much this technique across the different benchmarks.
>
> We tested 12 different workloads in this paper. The prefix cache hit rates for these workloads are shown in Figure 12 in Appendix C (L274 in the original paper). The cache hit rates range from 50% to 99%, indicating that the shared prefix is very common.
>
> Additionally, SGLang has been deployed in Chatbot Arena for real-world online serving. We observed a cache hit rate of more than 50% (see L295 in the original paper). This is because, in a multi-turn chat, the chat history serves as an accumulative prefix. Every new turn triggers a cache hit. (see also Figure 8)
>
> > How do you select the least recently used cache entry?
>
> We keep a timestamp on every tree node. The timestamp is updated every time that node is accessed. During eviction, we sort the leaf nodes based on this timestamp.
>
> > I wonder whether your scheduling strategy may lead to starvation.
>
> The scheduling strategy can lead to starvation. In L188 of the original paper, we mentioned this limitation. This limitation can be addressed by integrating simple lottery scheduling or more advanced fair scheduling [42].
>
> > I missed a description of the absolute times. One thing is to speedup 1 ms. and another 1000 days.
>
> Each benchmark in this paper typically takes several minutes to half an hour. This is because we mostly pick small datasets for faster experiments. The speedup will be 100% transferable to larger datasets. We will include a table showing the absolute time in the next version.
>
> The relative speedup is more relevant. For example, OpenAI runs the ChatGPT service, which can cost $700k per day (https://www.semianalysis.com/p/the-inference-cost-of-search-disruption). A small relative gain will translate to a huge absolute gain.
>
> > Finally you mention speeding for paralleillsim. One other important consideration is cost.
>
> Yes, using more GPUs can achieve lower latency at a higher cost. We mention parallelism to show that our technique is fully compatible with all parallelism strategies (data parallelism and tensor parallelism). Users can choose which parallelism to use based on their cost and latency requirements. Sometimes, adding more parallelism can be even more cost-efficient because it resolves some bottlenecks in the system (e.g., more GPU memory -> larger batch size -> more reuse).
>
> ---
>
> Please let us know if the answer addresses the weakness. We would greatly appreciate it if you could raise your score if you find the answers helpful.

---

> > ### Comment · Reviewer_P67b · 2024-08-13
> >
> > Thanks for the clear replies!

---

### Official Review · Reviewer_QRuS · 2024-07-19

**Soundness:** 4
**Presentation:** 4
**Contribution:** 4
**Rating:** 8
**Confidence:** 4

**Summary:**

This paper introduces SGLang, a comprehensive framework designed to enhance frontend LLM-based prompting/programming and to accelerate the runtime inference of LLM programs. In the frontend component, SGLang employs a meticulously structured primitive to streamline LLM prompting and support parallel execution. For the runtime component, the framework incorporates radix attention to optimize KV-cache reuse, implements LRU eviction of nodes in radix tree, and proposes more effective decoding with compressed finite state machine. This runtime significantly improves LLM inference efficiency. Experimental evaluations conducted on diverse LLM models across multiple downstream decoding tasks demonstrate the superior performance of SGLang.

**Strengths:**

- Great writing and presentation.
- Its proposed framework of formulating LLM-based prompting/programming in the frontend， and redesigning KV-cache as radix attention in the backend is neat and elegant. It is a significant contribution to LLM serving.
- The evaluation is extensive and solid. The results compared with powerful inference acceleration are very promising.

**Weaknesses:**

N/A

**Questions:**

This paper presents a comprehensive framework for the formulation of LLM prompting/programming and the acceleration of inference processes. The introduced method is both impressive and elegant, demonstrating extensive evaluations on both open-sourced and close-sourced LLMs that appear highly promising. This paper constitutes a substantial and significant contribution to the LLM serving community. I have some particular questions of the design inspirations and the detailed insights presented in the implementation.
- For the frontend component, the selection of primitives such as [gen], [fork], and [select] for building language programs is intriguing. I am interested in the inspiration of selecting these operations, like they are considered atomic and any language program can be represented through combinations of these primitives?
- The framework's implementation of an LRU eviction policy for managing nodes within the radix tree, which maps tokens to KV-cache tensors, is notable. However, I am wondering about the performance of alternative eviction policies, such as LFU.
- Regarding the inference of close-sourced LLM endpoints, such as GPT-4, the paper mentions that it facilitates inference through speculative decoding. However, I have the question that how this can be implemented in detail, like it is based on streaming and setting stop points during generation or else?

---

> ### Author Rebuttal · Authors · 2024-08-07
>
> We appreciate your valuable feedback. We will incorporate the clarifications and address the issues in the next draft. Here is our response to your questions:
>
> > For the frontend component, the selection of primitives such as [gen], [fork], and [select] for building language programs is intriguing. I am interested in the inspiration of selecting these operations, like they are considered atomic and any language program can be represented through combinations of these primitives?
>
> We select these primitive operations by summarizing the typical usage of LLMs. We summarize the usage patterns in the [OpenAI prompt engineering guide](https://platform.openai.com/docs/guides/prompt-engineering), the [Anthropic prompt engineering guide](https://docs.anthropic.com/en/docs/build-with-claude/prompt-engineering/overview), and more than ten papers on prompt design.
>
> It cannot be guaranteed that any language program can be represented by combining these primitives. However, SGLang can be easily extended to support new primitives. In addition, there are low-level APIs in SGLang that can access the probabilities of every token, which can theoretically do almost anything. These APIs are not introduced in this paper, but they can be useful when existing primitives are insufficient.
>
> > The framework's implementation of an LRU eviction policy for managing nodes within the radix tree, which maps tokens to KV-cache tensors, is notable. However, I am wondering about the performance of alternative eviction policies, such as LFU.
>
> The effectiveness of an eviction policy highly depends on the workload. We can easily support the LFU eviction policy as well. We compare LRU and LFU on our benchmarks: MMLU, Tree-of-Thought, and multi-turn chat. We find that LRU performs slightly better because our workload exhibits better temporal locality. We will add more experiments in the next version.
>
> > Regarding the inference of close-sourced LLM endpoints, such as GPT-4, the paper mentions that it facilitates inference through speculative decoding. However, I have the question that how this can be implemented in detail, like it is based on streaming and setting stop points during generation or else?
>
> It is implemented by getting the output chunk by chunk. It is not exactly streaming. For example, we set a parameter `num_api_spec_tokens`, which defines the number of speculative tokens. If this number is 64, we always request 64 tokens from the API endpoint and cache them. Then we match on these 64 tokens. If the speculation is correct, all 64 tokens will be used.

---

> > ### Comment · Reviewer_QRuS · 2024-08-12
> >
> > Thank the authors for the rebuttals. I am looking forward to the next version with more implementation details on API endpoints.

---

### Decision · Program_Chairs · 2024-09-25

**Decision:**

Accept (poster)

**Comment:**

Companies around the world are currently racing to apply LLMs to various problems and tasks.  Applying an LLM to a complex task typically involves interleaving requests/prompts to the LLM with code that processes results, and then issues new prompts.  In many cases, (e.g. tool use), the LLM must produce structured outputs (e.g. API calls).

SGLang is a library/domain-specific language that simplifies many of the common issues that arise when using an LLM in this way.  It can issue queries sequentially or in parallel, re-use KV-caches (to avoid pre-fill costs), and place structural constraints on the output of the LLM.  Most importantly, the authors have implemented the library using a number of LLM back-ends, including both open-source LLMs, and commercial LLMs that are only available through an API.

The reviewers unanimously recommend "accept" for this paper, and I concur.  As LLMs are incorporated into various products moving forward, libraries such as this one will become increasingly important.   SGLang may not be the final word on "how to we make LLMs easy to use for complex problems", but it seems like a valuable addition to the discussion.  The contribution of this paper is both timely and widely-applicable.

I personally would have liked to see more discussion around the design space for SGLang.  What are the pain points for writing LLM-programs in general, and what problems are _not_ solved by SGLang?  For example, the authors use a finite-state-machine for constrained decoding, but the purpose mainly seems to be efficiency.  For tool use, one might like to apply stronger constraints, e.g. to ensure that the output is syntactically correct (which requires more than an FSM), and possibly even type-checks.